# XRD Evaluation of Wurtzite Phase in MBE Grown Self-Catalyzed GaP Nanowires

**DOI:** 10.3390/nano11040960

**Published:** 2021-04-09

**Authors:** Olga Yu. Koval, Vladimir V. Fedorov, Alexey D. Bolshakov, Igor E. Eliseev, Sergey V. Fedina, Georgiy A. Sapunov, Stanislav A. Udovenko, Liliia N. Dvoretckaia, Demid A. Kirilenko, Roman G. Burkovsky, Ivan S. Mukhin

**Affiliations:** 1Nanotechnology Research and Education Centre of the Russian Academy of Sciences, Alferov University, Khlopina 8/3, 194021 Saint Petersburg, Russia; burunduk.uk@gmail.com (V.V.F.); bolshakov@live.com (A.D.B.); eliseevie@gmail.com (I.E.E.); fedina.serg@yandex.ru (S.V.F.); sapunovgeorgiy@gmail.com (G.A.S.); liliyabutler@gmail.com (L.N.D.); imukhin@yandex.ru (I.S.M.); 2Institute of Physics, Nanotechnology and Telecommunications, Peter the Great Saint Petersburg Polytechnic University, Politekhnicheskaya 29, 195251 Saint Petersburg, Russia; s_udovenko@mail.ru (S.A.U.); roman.burkovsky@gmail.com (R.G.B.); 3School of Photonics, ITMO University, Kronverksky Prospekt 49, 197101 Saint Petersburg, Russia; 4Ioffe Institute, Politekhnicheskaya 26, 194021 Saint Petersburg, Russia; zumsisai@gmail.com

**Keywords:** GaP, nanowire, wurtzite, zincblende, XRD, Rietveld refinement, TEM, molecular-beam epitaxy

## Abstract

Control and analysis of the crystal phase in semiconductor nanowires are of high importance due to the new possibilities for strain and band gap engineering for advanced nanoelectronic and nanophotonic devices. In this letter, we report the growth of the self-catalyzed GaP nanowires with a high concentration of wurtzite phase by molecular beam epitaxy on Si (111) and investigate their crystallinity. Varying the growth temperature and V/III flux ratio, we obtained wurtzite polytype segments with thicknesses in the range from several tens to 500 nm, which demonstrates the high potential of the phase bandgap engineering with highly crystalline self-catalyzed phosphide nanowires. The formation of rotational twins and wurtzite polymorph in vertical nanowires was observed through complex approach based on transmission electron microscopy, powder X-ray diffraction, and reciprocal space mapping. The phase composition, volume fraction of the crystalline phases, and wurtzite GaP lattice parameters were analyzed for the nanowires detached from the substrate. It is shown that the wurtzite phase formation occurs only in the vertically-oriented nanowires during vapor-liquid-solid growth, while the wurtzite phase is absent in GaP islands parasitically grown via the vapor-solid mechanism. The proposed approach can be used for the quantitative evaluation of the mean volume fraction of polytypic phase segments in heterostructured nanowires that are highly desirable for the optimization of growth technologies.

## 1. Introduction

Crystal phase engineering in semiconductor nanostructures provides a new design strategy for optoelectronic functional materials with a desired band structure. This can significantly improve the device performance and functionality produced by the absence of the heterointerface interdiffusion problem and consecutive sharp crystal phase boundaries in crystal-phase heterostructures [1,2]. 

Electronic properties of III-V semiconductor materials can vary significantly over the crystal structure, and a striking example is gallium phosphide (GaP). Under normal conditions, GaP exhibits a cubic zincblende (ZB) crystal structure with an indirect band gap of 2.26 eV [3,4,5]. Despite the broad optical transparency range and good lattice match with Si, the functionality of GaP-based optoelectronic devices is limited by high non-radiative recombination losses [6,7]. Recently, h, it was theoretically predicted and then, shortly after, demonstrated experimentally that GaP becomes a pseudo-direct bandgap material in hexagonal wurtzite (WZ) crystal phase [8,9,10].

Before, controlled stabilization of the GaP WZ phase was demonstrated only in gold- and silver- catalyzed epitaxial nanowires (NWs) [11,12,13,14,15]. But the use of Au catalyst limits NW-based device performance as it can act as a non-radiative recombination center [9,16,17,18], while Ag alloying of GaP could greatly modify the GaP band structure [15]. Metal-organic chemical vapor deposition (MOCVD) is commonly used to grow III-phosphide NWs, where carbon is inherently present and can incorporate impurities that affect the material and optoelectronic properties [19]. 

Thus, self-catalytic vapor-liquid-solid (VLS) NW growth via molecular beam epitaxy (MBE) technique is preferential over MOCVD au-catalyzed NWs growth. However, MBE-grown self-catalyzed GaP NWs commonly exhibit ZB crystal structure with random formation of stacking faults or rotational twins [4]. To the best of the authors’ knowledge, there are no reports on the GaP WZ-phase stabilization in self-catalytic NWs possessing a low density of impurity defect. Which makes them promising for future nanoelectronics and nanophotonics applications, where high structural quality is required. It was shown [20,21] that strain effects can have an essential influence on the electronic structure of crystal phase heterostructures. Thus, evaluation of the GaP WZ-phase lattice parameter is important to estimate strain at ZB/WZ crystal phase interface. One can also note in regard to the GaP WZ-phase, there are currently only a few reports on the GaP WZ lattice parameter [11,14].

Structural analysis of mixed crystal phase NWs is mostly limited to time-consuming electron microscopy techniques which lack assessment of integral information for the large area samples and do not allow evaluation of lattice parameters with high accuracy. At the same time, quantitative X-ray diffraction (XRD) assessment of the phase composition in lattice-matched nanoheterostructures, such as GaP NW arrays on lattice-matched Si substrate is not available due to overlapping of GaP ZB phase and Si diamond-like lattice Bragg peaks [22,23]. In addition, synchrotron-based light-sources are commonly used to perform XRD studies of epitaxial nanoheterostructures with a high-intensity dynamic range [11].

In this paper, we propose a complex approach unveiling the influence of the growth parameters on the crystal structure of self-catalytic MBE-grown GaP NWs. This approach is based on transmission electron microscopy (TEM)and powder XRD and reciprocal space mapping (RSM) using an in-house X-ray source. The XRD-RSM analysis allows us to study the polytypic phase inclusions and growth twins’ formation. In order to perform quantitative phase analysis and evaluate the WZ GaP phase lattice parameters, we present a simple express Rietveld refinement [24] approach based on acquiring powder diffraction patterns from the NWs exfoliated from a Si substrate.

## 2. Materials and Methods 

The experimental section is organized in the order of sample processing and study. First, in Section 2.1 we describe the epitaxial growth of NW arrays. Afterward, the methods of structural characterization of synthesized NW are presented (Section 2.2). The X-ray diffraction methods are listed in Section 2.3.

### 2.1. Synthesis of GaP NW Arrays

GaP NW arrays were synthesized on vicinal silicon (111) substrates with a 4° miscut in the <11$\bar{2}$> direction, using a Veeco GEN-III MBE machine equipped with Ga effusion cell and valved phosphorus cracker for P_2_ molecular flux (Tcracker = 900 °C). Silicon wafers were cleaned according to the modified Shiraki technique [25], which provides the formation of thin surface oxide in a boiling (110 °C) 35% dilute solution of nitric acid and deionized water. The boiling time was 10 min. Substrates were thermally degassed in ultrahigh vacuum conditions at 400 °C and then prior to the growth are thermally annealed for 30 min at the temperature 20 °C lower than the silicon oxide decomposition point (800 °C). This annealing procedure promotes defect formation in the surface oxide layer which acts as nucleation centers for further catalytic Ga droplet formation and was found to be crucial for vertical NWs growth with high surface density [26]. In order to prevent the catalytic droplet consumption at the end of the synthesis, the growth was interrupted by simultaneously shutting both groups-III and -V molecular fluxes followed by a turn of the substrate holder with its front-side from the molecular sources towards the vacuum pumps and cooling down by turning off the heating power.

### 2.2. Structural Characterization

The morphology of the synthesized NWs is studied using a scanning electron microscopy (SEM) (Zeiss SUPRA 25) (D-73446, Oberkochen, Germany). The values of the accelerating voltage and beam current were 20 kV and 300 pA, respectively. A crystal structure was studied with transmission electron microscopy (TEM) using a JEOL JEM–2100F field emission gun TEM (Tokyo, Japan) operating at 200 kV (with a point-to-point resolution of 0.19 nm in TEM mode). For TEM studies, NWs are “dry-transferred” to a carbon-coated TEM grid by sliding the grown sample on a TEM grid face to face.

### 2.3. X-ray Diffraction Studies

XRD reciprocal space mapping (RSM) was performed using Bruker KAPPA APEX DUO (Ettlingen, Germany) diffractometer equipped with microfocus Incoatec IµS X-ray source. The measurements are performed with Mo Kα-radiation (λ = 0.71 Å, 17.4 keV) with the 2D CCD X-ray detector. Semi-spherical cross-sections of the reciprocal space intensity distribution are reconstructed via mapping of the obtained frames into an intensity function in reciprocal space using a coordinate transformation derived from an Ewald sphere construction using the RecSpaceQT by Sergey Suturin 2019-07-25 (Saint Petersburg, Russia) software [27,28,29]. The sample was mounted with the surface normal oriented along the diffractometer phi axis, and a three-dimensional large-area reciprocal space map was obtained by performing a single 360° phi-scan (azimuthal rotation) with an angular step size ∆φ of 0.5°. Cylindrically shaped reciprocal space area bounded by the following range of scattering vectors Q: 0 Å < Q⊥ < 1.3 Å and −0.9 Å < Q‖ < 0.9 Å was obtained due to the fixed incident angle conical area of reciprocal space in the region of symmetric scattering vectors, Q > 0.5 Å was unattainable. To increase surface sensitivity, phi-scans are obtained with a fixed glancing angle at a grazing incidence condition (5° with respect to the surface).

The phase content was evaluated using powder X-ray diffraction. The study is performed using SuperNova (Rigaku Oxford Diffraction, Japan) diffractometer, radiation generated by IμS micro-focus X-ray tube with Cu Kα-radiation (λ = 1.5418 Å, ~8 keV) with a 2D CCD X-ray detector and measured in Debye-Scherrer geometry. Data are collected at two detector angle positions (0° and 60°). The average phase fraction is calculated with the use of the Rietveld refinement method [24,30,31,32,33] implemented in the FullProf (Winplotr) software package [34]. Rietveld refinement is a strong and simple method for determining crystal phase content in polyphase nanomaterials with different syngony [35,36,37,38]. All the powder patterns were refined using the Thompson–Cox–Hastings Pseudo Voigt functions.

## 3. Results and Discussion

### 3.1. NW Morphology and Structure

To study the influence of V/III flux ratio on NW crystal structure, samples are grown in a two stage approach with a sequential V/III flux ratio switch. Thus, NWs consist of two segments grown at equal growth time, but at different V/III flux ratios caused by the group-V flux variation. Group-III flux was kept constant during the NW growth and was equivalent to the 200 nm/h growth rate of planar GaP/Si(001), corresponding to the Ga beam equivalent pressure (BEP) of 7 × 10^−8^ Torr. According to planar GaP growth, we established the stoichiometric V/III flux ratio of 6. In all of the growth runs, at the first stage P_2_ BEP was set at 8.4 × 10^−7^ Torr (V/III ratio of 12) to obtain stable self-catalytic VLS NW formation reported previously [4,39]. At the second stage, P_2_ flux was increased up to 1.26 × 10^−6^ Torr (V/III ratio of 18) to promote gradual catalytic droplet consumption, which seems the most prominent way to control the catalyst droplet shape and thus the nucleation of the WZ phase [40]. A thin GaP_0.5_As_0.5_ marker segment with a thickness of ~20 nm was grown between these steps using the procedure reported previously [4]. The GaPAs markers allow us to distinguish the NW GaP segments grown at different V/III ratios. Schematic representation of the two-staged NWs and close-up SEM image of GaPAs markers for one of the grown samples is presented in Figure 1a inset. To demonstrate the influence of the growth temperature on NW crystal structure, three Samples were grown at the substrate temperature of 590 °C (Sample 1), 610 °C (Sample 2), and 620 °C (Sample 3). The growth temperature was measured by both thermocouple and pyrometer.

Typical cross-section SEM-images of the synthesized samples are presented in Figure 1b–d. One can note that vertical VLS NWs growth occurs simultaneously with the unwanted three-dimensional islands formed via vapor-solid (VS) mechanism [39,41]. Parasitic 3D islands are depicted with magenta polygons in Figure 1 and possess irregular shapes which can be associated with a large number of structural defects, as will be shown further in the XRD section. One can expect that NW array morphology, namely NW length and diameter (see Appendix A in Appendix A) should depend on the growth temperature. NWs of all the grown samples possess a hexagonal cross-section and are slightly tapered with a gradual diameter reduction towards the NW tip. Notably, NWs exhibit the same mean diameter value (120 ± 25.4 nm) at their tips independently on the growth temperature while the tapering angle reduces with the growth temperature decrease, i.e., the low-temperature sample exhibits a more enlarged NW base, which can be related to the radial vs. growth. 

It was observed that the catalytic droplets are absent on most NW tips, which indicates its consumption at the second growth stage at high group-V fluxes. The mean NW height exhibits non-monotonic dependence on the growth temperature and corresponds to 3.5 ± 0.9 μm, 4.2 ± 0.9 μm, and 3.6 ± 0.9 μm for Sample (1–3), respectively. The presence of the GaPAs marker allows to estimate the length of the upper and bottom NW segments grown at different V/III ratio and thereby evaluate the NW axial growth rate depending on the V/III ratio at various temperatures. Regardless of the sample, the dispersion Δh of bottom NW segment length his rather low (Δh/h~2–5%). Indeed, one can note that within the sample the GaPAs marker is found at the same height for each of the Samples (see Figure 1a. However, the length of the bottom NW segment depends on the growth temperature. On the contrary, the length of the upper NW segment suffers from high size dispersion (Δh/h~40–60%). It is evident that after the Ga droplet consumption at the second growth stage, the axial NW growth rate should drastically decrease, so dispersion of the NW upper segment length indicates that catalytic droplet consumption occurs non-simultaneously and is sensitive to the arrangement and dimensions of NWs. 

It can be noted that the length of the top NW segment increased with temperature and was the highest in Sample 3, which indicates the more rapid catalytic droplet consumption and termination of VLS axial growth at lower temperatures in Samples 1 and 2. It is known that both Ga and P adatom desorption increases with growth temperature raise, which is confirmed by lower axial NW growth rate. We assume that the main source of group-V species required for NW growth is a direct impingement of P_2_ molecules in the catalytic droplet since at the chosen growth temperatures P adatom diffusion is negligible [42]. At the same time, we assume that the increase in Ga adatom diffusion along the NW side facets with temperature leads to a decrease in the effective V/III ratio of atoms that can reach the droplet. This results in a slower Ga droplet consumption and an increase in the duration of the VLS growth stage.

Notably, there were no abrupt changes in the NW morphology associated with the increase in V/III ratio by 1.5 times and subsequent catalytic droplet consumption at the second NW growth stage—NW tapering angle mostly keeps constant along the NW length. This indicates that the interface area between the catalytic droplet and the GaP NW top facet, which determines the NW diameter, remains constant despite the catalytic droplet consumption.

### 3.2. TEM Study

Dark-field TEM (DF-TEM) imaging with a diffraction contrast was carried out to investigate the crystal structure of individual NWs. NWs were oriented with [110] ZB zone axis along the e-beam, which allows stacking faults and lattice twining to be distinguished. A typical DF-TEM image of a single NW of Sample 2 with a diffraction contrast corresponding to GaP WZ-phase is presented in Figure 2a,b (bright contrast corresponds to the WZ phase). It can be seen that stabilization of the WZ-phase occurs at the top and the base of NW, which is assumed to be associated with the droplet consumption and droplet size stabilization stages, correspondingly. The contrast between GaPAs and ZB-GaP segments is also visible in DF-TEM images (the GaPAs segment is marked in Figure 2a with the white arrow) as thin GaPAs nanodisks embedded in GaP NWs tend to stabilize in the WZ-phase due to the strain effects, as reported in our previous works [4,43]. The close-up DF-TEM images presented in Figure 2b reveal high-density lamellar twinning in the top NW segments with an average twin thickness of 7–8 nm. Note, the defect density is higher at the topmost NW part. Electron microdiffraction patterns acquired at the regions of NW with ZB and WZ structures are shown in Figure 2c,d, respectively. One can see a typical <110> zone axis electron diffraction pattern from a twinned ZB structure with a 180^°^ rotational twining around [111] NW growth direction [44,45]. Diffuse streaks oriented along the NW growth direction axis in the ZB diffraction pattern point out the high density of twin planar defects [14]. On the contrary, the WZ diffraction pattern demonstrates sharp spots, which indicates a low density of stacking faults.

As can be seen in Figure 2b, when the Ga droplet shrinks and the area of the catalytic droplet/NW interface decreases, the phase switches from WZ back to ZB. We assume that the WZ-phase formation in the upper part of the NW is associated with the stabilization of a certain range of contact angles of the catalytic droplet during its consumption [46,47]. Presumably, these angles allow GaP nucleation at the VLS triple-phase line favoring the WZ-phase stabilization via lower facet energies of vertical NW sidewalls [2,46]. Thus, ZB-phase stabilization can become again favorable during the formation of NW tip with inclined facets under shrinking catalytic droplets.

Analysis of DF-TEM images for the samples grown at different temperatures (see Figure 3a–c shows that the longest WZ-phase segments up to 500 nm in length are observed in Sample 2. The typical length of WZ insets is 163 ± 50 nm, 493 ± 96 nm, and 70 ± 20 nm for Sample 1–3, respectively. No correlation was found between the WZ segment length and the NW diameter. It should be noted that TEM analysis provides information only for several individual NWs, which makes it difficult to evaluate phase composition quantitatively. For this reason, we provide integral XRD crystal phase analysis in the next section.

### 3.3. XRD Study

#### 3.3.1. XRD Reciprocal Space Mapping

To perform integral analysis of the sample phase composition and crystal structure, reciprocal space maps (RSM) were recorded for the as-grown samples. ZB and WZ GaP reciprocal lattice node positions were modeled, to perform analysis of the experimentally acquired map. The detailed analysis of the reciprocal space cross-sections allows us to identify GaP epitaxial orientation with respect to Si (diamond structure), as well as twinning defect formation mechanisms and stabilization of ZB and WZ phases of GaP NWs. Two-dimensional RSM cross-section along [11$\bar{2}$] and [111]Si lattice directions acquired for Sample 2 is presented in Figure 4. For detailed analysis, we performed a numerical simulation, which helped to color the experimentally acquired map. Notably, the obtained map demonstrates similarity to the reflection high-energy electron diffraction (RHEED) pattern taken along [1$\bar{1}$0] azimuth observed *in-situ* during the MBE growth (see in Appendix A in Appendix A). Note that the direct beam, marked as 000 spot, is also visible, due to the relatively low absorption of Mo Kα-radiation in Si (absorption length ~650 µm) [48]. The brightest spots corresponding to the Si[$\bar{1}$10] zone axis pattern, with the corresponding model of the Si[1$\bar{1}$0] zone diffraction pattern, are depicted by orange dots in Figure 4. As discussed in the experimental section, vicinal Si substrates with a 4° miscut towards [11$\bar{2}$] are used, so the observed zone axis patterns are tilted relative to the substrate normal. Since the substrate diffraction peaks have high intensity, their instrumental broadening along the axis directed towards the origin (*hkl* = 000) of the reciprocal lattice is observed. This phenomenon is caused by the beam divergence and elongated footprint at a shallow incidence angles.

Despite insufficient lattice mismatch between Si and ZB GaP, the coincident orientation of ZB GaP and Si lattices can be distinguished by the appearance of a ZB-GaP 002 Bragg peak in its [1$\bar{1}$0] zone axis pattern, which is basis-forbidden for Si. Corresponding model positions of ZB-GaP diffraction peaks are depicted with green circles in Figure 4a. The second set of the ZB-GaP diffraction spots sharing the same row, corresponding to the rotation of the GaP lattice by 180° along the [111] growth direction, can also be distinguished. [$\bar{1}$10] zone axis pattern model of the [111] growth twin is depicted by dashed blue circles in Figure 4a, while the gradient-colored arrow illustrates the lattice rotation axis. Thus, the observed twin diffraction pattern corresponds to the multiple lamellar (111) twins, also confirmed by TEM.

One can also note the appearance of the additional set of the off-specular diffraction spots located in the middle of split pairs of the rotationally twinned ZB-GaP and Si Bragg peaks. The observed diffraction spots correspond to the WZ GaP [11$\bar{2}$0] zone axis pattern. The corresponding model of the WZ zone axis pattern is depicted by the red circles in Figure 4a. The cross-section of the RSM through the off-specular 11$\bar{1}$ and 220 Si Bragg peaks along the [111] direction demonstrating relative position of the ZB and WZ GaP reflexes is shown in Figure 4b The acquired profile shows the presence of both Si and twinned ZB-GaP peaks with WZ phase Bragg peaks in between, at integer and half-integer value positions of normalized perpendicular component of diffraction vector |Q⟂|/|Q_111_GaP| [11]. This observation also coincides with TEM results.

It should be taken into account that both NWs and parasitically grown islands make a contribution to the acquired diffraction pattern. Formation of the rotational twins in the ZB lattice is a common effect if the growth facet is one of the {111} planes [49], and thus three-dimensional islands with {111} facets can also possess rotational twinning. To untangle all possible GaP lattice orientations, a spherical cross-section of a fixed-length X-ray scattering vector was cut from the obtained RSM data and its stereographic projection (pole-figure) was plotted. The obtained pole figures for the scattering vector values corresponding to the diffraction from ZB {111} and WZ {1$\bar{1}$02} crystal planes are presented in Figure 5 and show all possible orientations of the chosen crystallographic planes.

Similarly, to the 2D-RSM cross-section in Figure 4, corresponding model positions of diffraction peaks for ZB-GaP lattice which is co-oriented with the Si substrate are depicted with green circles. The presence of additional first-order rotational twins related to 180° lattice rotation along three possible inclined <11$\bar{1}$> ZB lattice directions are clearly seen from the obtained pole figure for the ZB-GaP {111} scattering vector presented in Figure 5a. For clarity, we mark first-order twinning only in the [111] and [11$\bar{1}$] directions. Corresponding zone axis pattern of the [11$\bar{1}$] rotational twin is marked by purple circles in Figure 4. As can be seen from the pole figure, two other twinned lattice orientations, namely [$\bar{1}$11] and [1$\bar{1}$1] growth twins, which [$\bar{1}$10] zones are lying outside the imaged in Figure 4a also appear. It should be noted that the observed distribution of {111} planes cannot be explained only by the first-order twinning. In turn, as can be seen from the pole figure in Figure 5a, three new possible {111} twining plane orientations emerged with each first-order rotational twinning along <11$\bar{1}$>, thus 9 second-order twins are present, however, for clarity in Figure 5 we mark second-order twinning only in [$\bar{1}\;\bar{1}\;\bar{1}$] and [1$\bar{1}\;\bar{1}$] directions. Their diffraction peaks are marked in pole figure in Figure 5 by thin blue and orange circles correspondingly. The [1$\bar{1}$0] zone axis patterns of the second-order [$\bar{1}\;\bar{1}\;\bar{1}$] rotational twin are fit into the [110] zone imaged in Figure 4a and marked by thin blue circles while the corresponding lattice rotation axes are marked by the gradient colored arrow. Since, according to TEM data, NWs exhibit rotational twinning only along the <111> growth direction, we assume that the second and third-order twinning has mainly occurred in parasitically grown islands.

One can also note that the observed diffraction pattern has 180-degree rotational symmetry about the [111] axis. One can conclude that the above-mentioned mechanisms are applicable to the [111] growth twin. As a result, 3 possible orientations of second and 9 possible orientations of third-order twins occur. Thus, all 26 = 2 + (3 + 9) × 2 crystallographic orientations can be distinguished from the provided RSM. On the contrary, only a single crystallographic orientation of the WZ lattice was found. The obtained pole figure presented in Figure 5b demonstrates only six characteristic 1$\bar{1}$02 WZ Bragg peaks, confirming a single possible orientation of WZ GaP lattice with a [0001] WZ crystal direction aligned along <111> growth direction of NWs. This observation confirms that WZ-phase formation occurs only in vertically oriented NWs. Thus, one can suggest that the NW array should be detached from the Si substrate with a parasitically grown GaP island layer that should be separated to quantitatively evaluate the mean volume fraction of the WZ phase in self-catalyzed GaP NW.

#### 3.3.2. Quantitative Phase Analysis

The acquired XRD curves can be approximated by a superposition of two diffraction patterns (see example in Figure 6b) corresponding to the cubic ZB GaP phase belonging to *F*$\bar{4}$*3m* space group [50,51], and the WZ phase belonging to *P6_3_mc* space group (see XRD pattern models for these crystal phases in Appendix A in Appendix A) [14,31,52]. The simulated XRD patterns in Figure 6b were extracted from FullProf (Winplotr) output data after Rietveld refinement for each crystal phase separately, where the red line—simulated XRD pattern for WZ phase, green line—simulated XRD pattern for ZB phase, and brown line—experimental data of Sample 2 after background subtraction. The corresponding Bragg positions are shown with green and red marks in Figure 6b, while the indexes of ZB and brightest WZ Bragg reflexes are labeled in Figure 6b, respectively. The Rietveld refinement results are presented in Table 1. The highest WZ volume content of 9.7% was observed for Sample 2, which well correlates with the obtained TEM data reported above. Appendix A in Appendix A shows the refined XRD pattern for Sample (3) with the lowest WZ content. The obtained data provide statistics across a large number of disordered NWs in one measurement, which makes it more practically useful compared to TEM analysis.

Despite the partial overlapping of ZB and WZ Bragg reflexes [14,53], the obtained experimental data allow refinement of the unit cell parameters of the WZ GaP phase. The refined *a* and *c* lattice parameters for the WZ unit cell are 3.839 ± 0.007Å and 6.34 ± 0.015 Å respectively and differ from a theoretically predicted on 1% [52] and are consistent with WZ lattice parameters reported by Assali et al., in [11]. Noteworthy (111)_ZB_ and (0001)_WZ_, as well as {110}_ZB_ and {11$\bar{2}$0}_WZ_ lattice plane distances, are different for the ZB and WZ lattices, indicating possible lattice stress at the WZ/ZB GaP interface. The refined parameter of the ZB structure equals 5.448 ± 0.003 Å.

The obtained results are consistent with TEM microstructural analysis and provide statistical data across a large number of NW in one measurement. Thus, the reported approach can be used as a simple express technique for quantitative evaluation of the NW phase composition.

## 4. Conclusions

We report on the controllable stabilization of the polymorphic wurtzite crystal phase in the MBE grown self-catalyzed GaP NWs by varying the V/III flux ratio and growth temperature, which is promising for future device implementation due to direct-to-indirect bandgap transition in WZ-GaP. A complex approach based on the DF-TEM and XRD-RSM and powder-XRD techniques were proposed for the analysis of the epitaxial GaP NW crystal structure. For the investigation of the NWs crystallinity we, first, employ TEM that enables direct observations of the WZ phase stabilization up to 500 nm long GaP NW segments within the self-catalytic approach.

For the investigation of the crystallinity of NW arrays, we use the combined XRD-based approach with the use of an in-house X-ray source allowing for statistical evaluation of the NWs phase composition. The obtained XRD powder diffraction data are used to refine the parameters of the metastable wurtzite GaP phase. The evaluated *a* and *c* lattice parameters of the GaP WZ unit cell are estimated as 3.839 ± 0.007 Å and 6.340 ± 0.015 Å, respectively. The highest WZ phase content of 9.7% is observed for the sample grown at 610 °C and V/III ratio of 18. X-ray diffraction reciprocal space mapping reveals that the WZ-phase formation occurs only in vertically oriented nanowires during VLS growth, while the WZ phase is absent in GaP islands parasitically grown via the vs. mechanism. Also, XRD analysis allows studying the formation of growth twins in NWs and parasitically grown islands in detail.

The results of the work open new perspectives for high phase purity phosphide NWs synthesis and its fast and reliable investigation with XRD techniques using an in-house X-ray source. Crystal phase engineered MBE grown self-catalyzed NWs are very promising for future nanoelectronics and nanophotonics applications, as they are free from the extrinsic impurity’s defects caused by the residual catalyst material used in the growth process affecting their optoelectronic properties.

## Figures and Tables

**Figure 1 nanomaterials-11-00960-f001:**
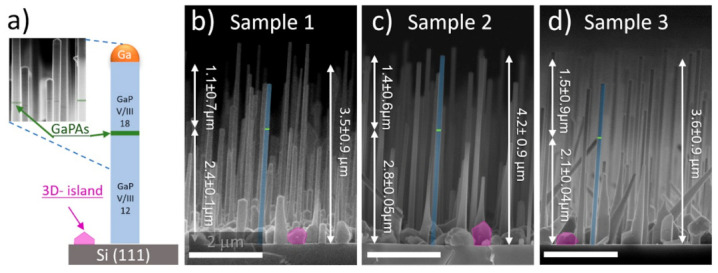
(**a**) schematic representation of the two-staged growth procedure; (**b**–**d**) Cross-sectional SEM-images of the studied GaP NW arrays.

**Figure 2 nanomaterials-11-00960-f002:**
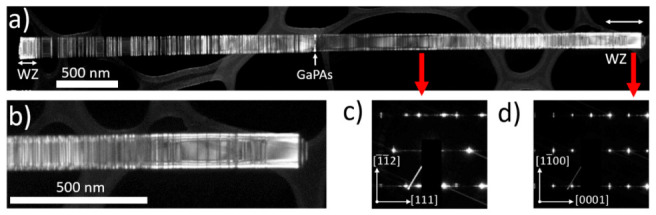
(**a**) dark-field TEM images obtained with WZ [$\bar{1}$100] reflection, (**b**) zoomed dark-field TEM image and corresponding electron diffraction patterns acquired from (**c**) ZB and (**d**) WZ GaP NW segments, taken along [$\bar{1}$ 10] [$\bar{1}$120] directions correspondingly. The inclined grey stripes appear due to the TEM CCD camera artifacts.

**Figure 3 nanomaterials-11-00960-f003:**
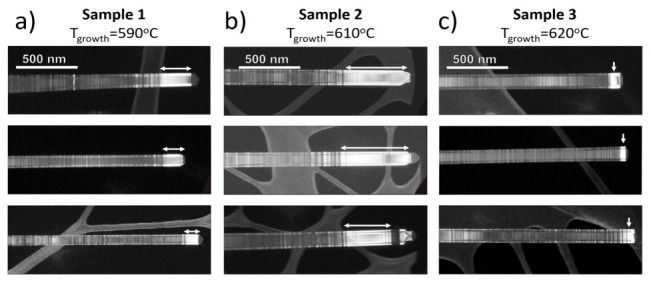
TEM images of NW tips of the studied samples (**a**) Sample 1, (**b**) Sample 2 and (**c**) Sample 3—WZ phase is depicted with white arrows.

**Figure 4 nanomaterials-11-00960-f004:**
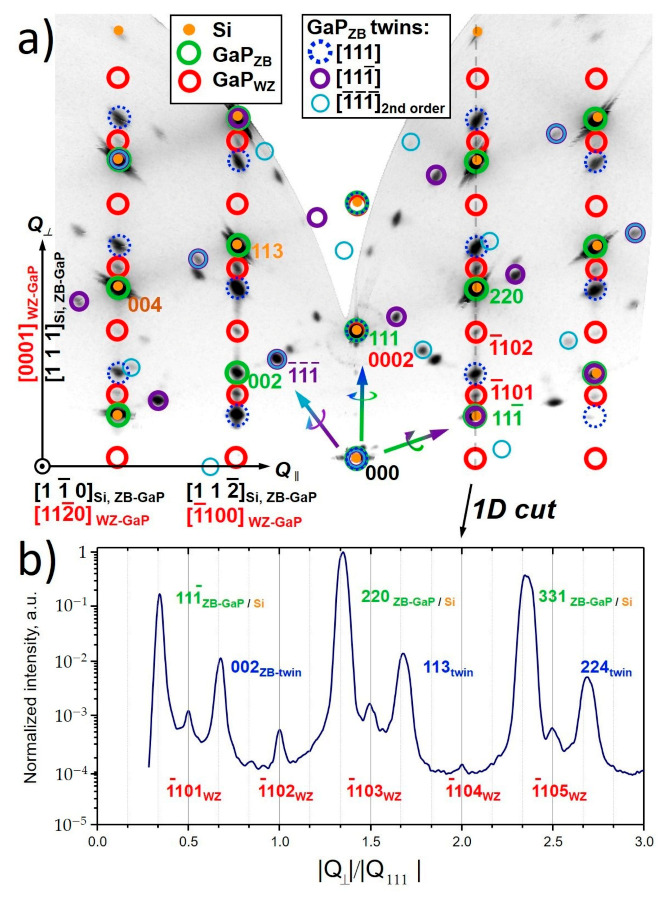
(**a**) Two dimensional XRD reciprocal space cut along [11$\bar{2}$]Si and [111]Si directions obtained for Sample 2 with superimposed models of Si[1$\bar{1}$0] (orange dots), ZB-GaP [1$\bar{1}$0] (green circles), ZB-GaP 1st (blue dashed and violet circles) and 2nd order rotational twins (turquoise circles) and WZ-GaP [11$\bar{2}$0] (red circles) zone axis patterns shown with coloured circles, (**b**) Line-cut of the RSM through the off-specular Si (220) diffraction reflex along the [111] direction, demonstrating the relative position of the ZB and WZ reflexes. Q⟂—perpendicular component of the diffraction vector.

**Figure 5 nanomaterials-11-00960-f005:**
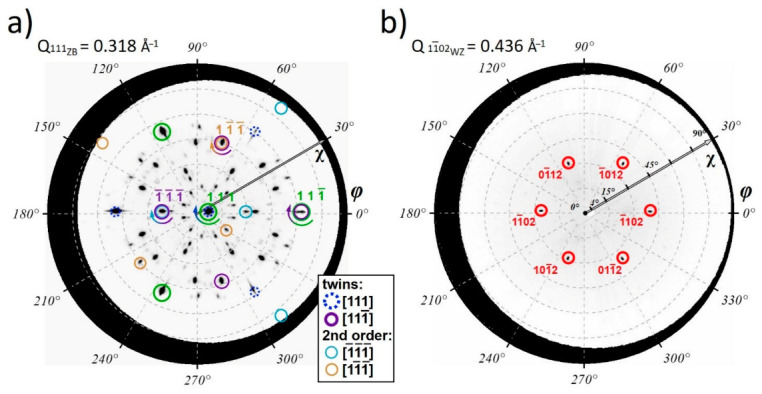
X-ray pole figure maps corresponding to (**a**) ZB-GaP {111} (d = 3.145 Å) and (**b**) WZ-GaP {1$\bar{1}$02} Bragg reflections (d = 2.294 Å). The modeled reflection positions are shown with coloured circles, gradient colored arrows indicate the formation of rotational twins in ZB-GaP.

**Figure 6 nanomaterials-11-00960-f006:**
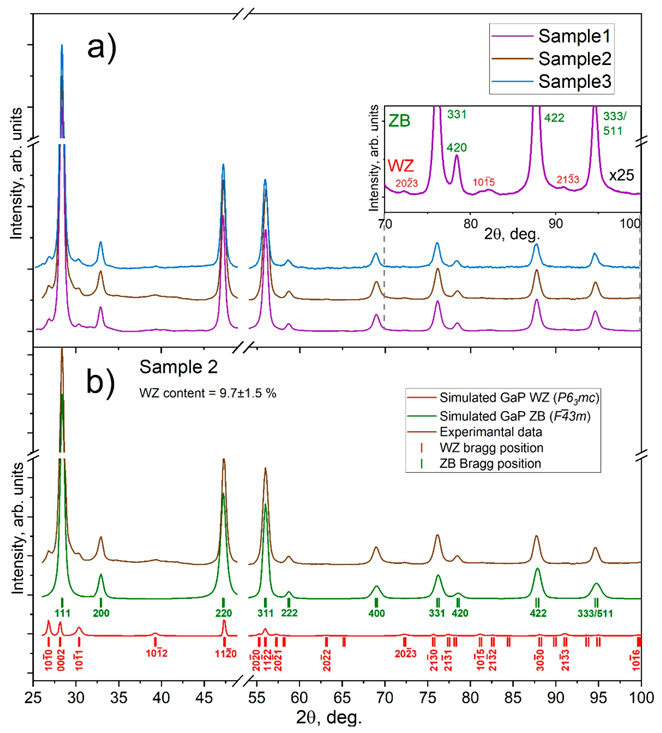
(**a**) X-ray powder diffraction patterns of GaP NW embedded in scotch adhesive, inset—zoomed area of XRD pattern, (**b**) XRD patterns of Sample 2, where: the red line is the profile of simulated XRD pattern for GaP WZ space group, the green line is a simulated profile of ZB XRD pattern and the brown line is experimental data after background subtraction, offset for clarifying.

**Table 1 nanomaterials-11-00960-t001:** Calculated crystal phase content in the investigated samples with the corresponding growth parameters.

Sample	Lattice Parameter *a* WZ, Å	Lattice Parameter *c* WZ, Å	WZ Content, %	T_growth_, ^°^C
Sample 1	3.840	6.349	~8.1 ± 0.93	590
Sample 2	3.839	6.344	~9.7 ± 1.5	610
Sample 3	3.839	6.345	~6.4 ± 1.14	620

## Data Availability

The data presented in this study are available on request from the corresponding author.

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
