# Peer review of "XRD Evaluation of Wurtzite Phase in MBE Grown Self-Catalyzed GaP Nanowires"

_nanomaterials, 2021, doi:10.3390/nano11040960_

Round 1

Reviewer 1 Report

In this paper by Olga Yu. Koval and co-workers report how a wurtzite phase in molecular beam epitaxy grown self-catalyzed GaP nanowires can be studied by XRD. The article was improved considerably after resubmission. It contains many interesting findings, which may be valuable to a wider audience. Two issues remain, which should be handled before the paper can be accepted. Please find them below:
1) Redundant empty space should be removed (e.g. Page 7). 
2) (Once again the same comment) "NWs exhibit the same mean diameter value (120±25,4 nm)" - for a material to be classified as a nanomaterial, it must be constrained to 100 nm in at least one dimension. This does not seem to be the case in this work, so the authors may reconsider the way these structures are called. Especially that the journal to which the paper is submitted is called Nanomaterials, so it is of utmost importance to ensure that a proper type of material is analyzed. Please comment on this issue. 

Author Response

In this paper by Olga Yu. Koval and co-workers report how a wurtzite phase in molecular beam epitaxy grown self-catalyzed GaP nanowires can be studied by XRD. The article was improved considerably after resubmission. It contains many interesting findings, which may be valuable to a wider audience. Two issues remain, which should be handled before the paper can be accepted. Please find them below:

1) Redundant empty space should be removed (e.g. Page 7).
answer:  The authors thank the Reviewer for carefully reading our manuscript. According to the comment, we have removed the redundant empty spaces throughout the whole text. 

2) (Once again the same comment) "NWs exhibit the same mean diameter value (120±25,4 nm)" - for a material to be classified as a nanomaterial, it must be constrained to 100 nm in at least one dimension. This does not seem to be the case in this work, so the authors may reconsider the way these structures are called. Especially that the journal to which the paper is submitted is called Nanomaterials, so it is of utmost importance to ensure that a proper type of material is analyzed. Please comment on this issue. 

Answer: We agree with the Reviewer that, from the classical point of view, nanostructures should be less than 100 nm in at least one dimension. However, this rule is not strong. For example, plasmonic and all-dielectric nanoantennas normally have dimensions exceeding several hundreds of nm (typical height and diameter are about 180 nm and  250 nm, respectively) [1],[2]. In the case of prolonged nanostructures, the length of nanoantennas can be more than several µm [3]. In this field, the term “optical nanoantenna” is valid until the size of nanostructures is less than the operating wavelength, which is around 550 nm for optical spectral range. The waveguides made of low loss material such as GaP, the sizes of which should be a little bit bigger than 100 nm in order to support fundamental optical modes, are also subject to nanophotonics [4]. 

Moreover, bottom-up-grown semiconductor nanowires (NWs) with a diameter slightly exceeding 100 nm are a prospective technological platform for future nano-optoelectronics including single-photon sources and detectors as they allow to form both axial and radial heterostructures including quantum dot and quantum well in NW structures [5].

In sub-µm scale structures with a high aspect ratio, such as quasi-1D NWs, a number of finite-size and surface effects related to metastable crystal phase stabilization, strain relaxation, and electromagnetic field localization can be observed [6]. 

In this work, we present the stabilization of the polymorphic wurtzite crystal phase in GaP NW and GaPAs nanodisks embedded in NWs understudy as one of the above-mentioned reduced dimensionality effects. Note, that the length of WZ phase segments in samples 3 and GaPAs nanodisks in all GaP NWs understudy is less than 100 nm. It is demonstrated that the procedure of express XRD analysis presented in the paper can be used for the quantitative evaluation of phase composition and volume of nano-sized NW segments with relatively high accuracy. Thus, we do assume that the material under study (nanowire heterostructure), as well as presented approach for approach below to the field of nanomaterials [7],[8].

[1] Alù, A., & Engheta, N. (2008). Tuning the scattering response of optical nanoantennas with nanocircuit loads. Nature Photonics, 2(5), 307–310. doi:10.1038/nphoton.2008.53 

[2] Krasnok, A. E., Maksymov, I. S., Denisyuk, A. I., Belov, P. A., Miroshnichenko, A. E., Simovski, C. R., & Kivshar, Y. S. (2013). Optical nanoantennas. Physics-Uspekhi, 56(6), 539–564. doi:10.3367/ufne.0183.201306a.0561 

[3] Moiseev, E. I., Kryzhanovskaya, N., Polubavkina, Y. S., Maximov, M. V., Kulagina, M. M., Zadiranov, Y. M., Zhukov, A. E. (2017). Light Outcoupling from Quantum Dot-Based Microdisk Laser via Plasmonic Nanoantenna. ACS Photonics, 4(2), 275–281. doi:10.1021/acsphotonics.6b00552 

[4] Aravind P. Anthur, Haizhong Zhang, Yuriy Akimov, Jun Rong Ong, Dmitry Kalashnikov, Arseniy I. Kuznetsov, and Leonid Krivitsky, Second harmonic generation in gallium phosphide nano-waveguides, Optics Express Vol. 29, Issue 7, pp. 10307-10320 (2021), doi:10.1364/OE.409758

[5] Leandro, L., Gunnarsson, C. P., Reznik, R., Jöns, K. D., Shtrom, I. V., Khrebtov, A. I., … Akopian, N. (2018). Nanowire quantum dots tuned to atomic resonances. Nano Letters. doi:10.1021/acs.nanolett.8b03363 

  [6] Lieber, C. M., & Wang, Z. L. (2007). Functional Nanowires. MRS Bulletin, 32(02), 99–108. doi:10.1557/mrs2007.41
[7] E. Barrigon, M. Heurlin, Z. Bi, B. Monemar, and Lars Samuelson, Synthesis and Applications of III−V Nanowires, Chem. Rev. 2019, 119, 15, 9170–9220, doi: 10.1021/acs.chemrev.9b00075

[8] LaPierre, R. R., Chia, A. C. E., Gibson, S. J., Haapamaki, C. M., Boulanger, J., Yee, R., … Rahman, K. M. A. (2013). III-V nanowire photovoltaics: Review of design for high efficiency. Physica Status Solidi (RRL) - Rapid Research Letters, 7(10), 815–830. doi:10.1002/pssr.201307109

Reviewer 2 Report

The manuscript corrected according reviewer marks is ok. Can be accepted

Author Response

We thank the Reviewer for the high evaluation of our manuscript.

Round 2

Reviewer 1 Report

Revisions completed satisfactorily. I recommend the publication of the article. 

This manuscript is a resubmission of an earlier submission. The following is a list of the peer review reports and author responses from that submission.

Round 1

Reviewer 1 Report

In this work by Olga Yu. Koval et al., the authors studied the growth of GaP nanowires (vide Comment #3). The results are generally sound and interesting, but there are some issues that must be tackled before the paper can be recommended for publication. The major concern is a possible mismatch between the theme of the manuscript and the aims and scope of the journal Nanomaterials. Please find the comments below:
1) A crucial requirement for a research paper is that it needs to be reproducible. However, this work lacks certain details, which does not enable other researchers to repeat it. Consequently, not only the claims cannot be validated, but the community cannot build on these findings to increase the impact of this contribution. Please carefully screen the experimental section to include all the parameters. Examples of shortcomings are given below:
- modified Shiraki technique should be cited
- there is no details regarding the time of exposure to boiling (110°C) 35% dilute solution of nitric acid and deionized water
- degassing/annealing parameters are not specified
- SEM acceleration voltage
- etc.
2) "." should be the decimal point.
3) "NWs exhibit the same mean diameter value (120±25,4 nm)" - for a material to be classified as a nanomaterial, it must be constrained to 100 nm in at least one dimension. This does not seem to be the case in this work, so the authors may reconsider the way these structures are called. Please comment on this issue. 
4) Conclusions section should be extended and include a future outlook.

Reviewer 2 Report

  1. how temperature was measured (subsection 2.1)?
  2. Please identify the size of a three-dimensional large-area RSM investigation (103 line)
  3. Why for detailed DF-TEM study was selected sample 2? Please describe the motivation of choice.
  4. 189 line Figure 2b and e. There is no image e on the Figure 2. Figures 2 c and d contains black squares and white and grey lines. Please clarify the role of those elements in the figure caption.
  5. Why for XRD RSM was selected sample 1? Large volume of WZ in sample 2 could allow to determine more precise information about WZ reflexes. Please explain the motivation of choice. (subsection 3.3). Could You compare the results of RSM obtained for sample 2 and 3. comparison is
  6. Subsection 3.3.2. lines 304-307. In order to determine the photoluminescence properties of the synthesized arrays of NWs and the parasitic layer separately, we studied the released membrane and as-grown NW array. After membrane exfoliation, SEM imaging was used to clarify how exactly NWs were encapsulated into polymer membrane (see Figure 5 f) and then were exfoliated from Si substrate (see Figure 5 d). The reviewer did not found both the results of PL investigation and the Fig 5. Please upload the investigation set up, conditions, spot area, and figure.
  7. section 3.3.2. Please describe how R-factor was calculated. The table 1 must be updated by lattice constants a and c for all samples. The discussion about relationship between R-factor, lattice constants and growth conditions must be included.
  8. The conclusions must be extended